# A Comparative Evaluation of Lumped and Semi-Distributed Conceptual Hydrological Models: Does Model Complexity Enhance Hydrograph Prediction?

**Emmanuel Okiria [1], Hiromu Okazawa [2], Keigo Noda [3,*], Yukimitsu Kobayashi [4], Shinji Suzuki [2] and Yuri Yamazaki [2]**

1   The United Graduate School of Agricultural Sciences, Gifu University, Gifu 501-1193, Japan; a6102003@edu.gifu-u.ac.jp
2   Faculty of Regional Environment Science, Tokyo University of Agriculture, Tokyo 156-8502, Japan; h1okazaw@nodai.ac.jp (H.O.); s4suzuki@nodai.ac.jp (S.S.); yy206792@nodai.ac.jp (Y.Y.)
3   Faculty of Applied Biological Sciences, Gifu University, Gifu 501-1193, Japan
4   NTC-International Co., Ltd., Tokyo 136-0071, Japan; y.kobayashi@ntc-i.co.jp
*   Correspondence: anod@gifu-u.ac.jp

**Abstract:** The prediction of hydrological phenomena using simpler hydrological models requires less computing power and input data compared to the more complex models. Ordinarily, a more complex, white-box model would be expected to have better predictive capabilities than a simple grey box or black-box model. But complexity may not necessarily translate to better prediction accuracy or might be unfeasible in data scarce areas or when computer power is limited. Therefore, the shift of hydrological science towards the more process-based models needs to be justified. To answer this, the paper compares 2 hydrological models: (a) the simpler tank model; and (b) the more complex TOPMODEL. More precisely, the difference in performance between tank model as a lumped model and the TOPMODEL concept as a semi-distributed model in Atari River catchment, in Eastern Uganda was conducted. The objectives were: (1) To calibrate tank model and TOPMODEL; (2) To validate tank model and TOPMODEL; and (3) To compare the performance of tank model and TOPMODEL. During calibration, both models exhibited equifinality, with many parameter sets equally likely to make acceptable hydrological simulations. In calibration, the tank model and TOPMODEL performances were close in terms of 'Nash-Sutcliffe efficiency' and 'RMSE-observations standard deviation ratio' indices. However, during the validation period, TOPMODEL performed much better than tank model. Owing to TOPMODEL's better performance during model validation, it was judged to be better suited for making runoff forecasts in Atari River catchment.

**Keywords:** lumped model; distributed model; semi-distributed model; tank model; TOPMODEL; equifinality





## 1. Introduction

The role of hydrological monitoring and modelling in the quantification of Ecosystem Service (ES) flows in Paid Ecosystem Service (PES) schemes cannot be overlooked [1,2]. In proposing that PES schemes be built around the ESs flowing from the operation of irrigation and drainage infrastructure, Okiria et al. [3] emphasised the role of hydrological models in such schemes. In addition, among other uses, hydro-meteorological data are indispensable in the design and operation of irrigation and drainage schemes, as well as other hydraulic infrastructure.

Ideally, hydrological data should be collected by detailed monitoring and observation of the catchment. However, detailed spatial measurements are prohibitively expensive and difficult to implement, and in unobserved catchments or in an unobservable future, data is unavailable [4]. Globally, catchment areas less than 2500 km$^2$ are overwhelmingly

unmonitored, with efforts to monitor them becoming increasingly difficult [5]. Consistent with these findings, in Uganda, the majority of the rivers are ungauged [6] and furthermore, some river flow rating curves were found to be erroneous and in need of updating [7]. In such situations, hydrological modelling can be used as a tool to generate estimates of hydrological data.

However, hydrological models are built around simplifying assumptions of natural hydrological systems [8]. In order from the simplest to the most complex, hydrological models can be divided into black-box (statistical), grey-box (conceptual) or white-box (physically-based) models. Black-box models lack a description of the underlying hydrological processes and are usually expressed as empirical models. Their grey-box counterparts attempt to describe the underlying physical processes and are based on empirical equations that are more complex than those seen in black-box models. The white-box model description is more strongly grounded in the physics of the underlying hydrological processes and has equations that are based on the laws of conservation of mass and energy [4].

Based on the smallest unit of hydrological similarity, hydrological models can also be categorised as lumped, semi-distributed or distributed. The simplest hydrological models by scale, the lumped models, assume homogeneity of hydrological response across the entire catchment. Meanwhile, the more complex distributed models consider the spatial variability of the catchment physical properties [9]. Semi-distributed models on the other hand are optimised to benefit from the strengths of both lumped and distributed models.

As hydrology advances towards more complex, process-based models [4], two questions arise: (1) Does more complexity translate into higher model prediction accuracy? (2) Within what limits can the simplest models be used? This question is important because: Simpler hydrological models have the benefits of ease of use, simplicity in interpretation by non-experts and low demand for computer processing power-a desirable situation. It is therefore important to confirm the limits within which they are applicable.

To clarify this, this paper compares the performance difference during model calibration and validation between the 'simpler' tank model as a lumped parameter model and the 'more complex' TOPMODEL concept as a semi-distributed model in a catchment in Eastern Uganda. In this study, the 'complexity' of a model was defined based on: (a) the input data need; (b) the scale of hydrological homogeneity and (c) the extent to which the model described physical hydrological processes. Compared to tank model, TOPMODEL required an extra layer of input data: topographic data. Tank model was lumped while TOPMODEL was semi-distributed. Additionally, the underlying equations of TOPMODEL are more physically-based than those of tank model.

The bulk of tank model application has been in sub-tropical, Eastern Asia e.g., in [10–14] to simulate catchment-scale rainfall-runoff responses. Researchers in Eastern Asia also used tank model to simulate the hydrological response of rice paddy fields [15–18]. Chikita et al. [19] tested the applicability of tank model to a subarctic catchment in Alaska. In Sugawara [20], a case for the physical meaning of tank model was argued when it was used to prove the existence of the separated storage of ground water. In another study, Hong et al. [21] used tank model to simulate ground water levels in Kumamoto, Japan.

However, literature on the application of tank model in tropical climates is scanty. In the Horn of Africa, Onyutha [22,23] applied it to the Blue Nile Basin in Ethiopia and Sudan. In East Africa, tank model was applied to catchments in both Rwanda [24] and Uganda [25,26]. More specifically, in Eastern Uganda, Okiria et al. [25] used it to simulate the daily discharge hydrograph of Atari River catchment while Mubialiwo et al. [26] simulated the flood discharge hydrograph of Malaba River catchment.

There is need for a systematic and extended application of tank model in tropical climates, with moderate humidity, to elucidate if the underlying model assumptions still hold in such conditions.

TOPMODEL has been widely applied in humid, temperate climates in Europe, especially in the United Kingdom, to simulate the rainfall-runoff process at catchment scale [27–29] and in China [30,31]. From these applications, the model was observed

to perform well in humid climates with wet and shallow soils. Abou-shanab et al. [32] tested the validity of the TOPMODEL assumptions in the complex climate and topography of Nepal, achieving reasonable runoff hydrograph simulations. There is also evidence of its application to Mediterranean climates [33,34].

Despite this, literature on the application of TOPMODEL in tropical climates, especially in Africa, is sparse. Some of the applications of TOPMODEL in Africa included in humid, tropical climates in West Africa [35,36] and in the Horn of Africa, Ethiopia [37,38]. In East Africa, Okiria et al. [39] used the TOPMODEL concept to predict the daily discharge hydrograph, and to identify the minimum number of rainfall events required to calibrate the model in Atari River catchment in Eastern Uganda.

Similarly, a systematic and expanded application of TOPMODEL in tropical climates, with moderate humidities, is necessary to confirm if the assumptions on which the model is built still hold.

The requirement for a systematic evaluation of the applicability of the 2 models, as well as a comparison of the difference in performance between tank model as a lumped parameter model and TOPMODEL as a semi-distributed model in Eastern Uganda is identified. Consequently, this study performed a comparative predictive performance analysis between tank model and TOMODEL in the Atari River catchment in Eastern Uganda. The objectives were: (1) To calibrate tank model and TOPMODEL; (2) To validate tank model and TOPMODEL; and (3) To compare the performance of tank model and TOPMODEL.

## 2. Materials and Methods

### 2.1. Study Area

The study area is the Atari River catchment in Eastern Uganda, with a drainage area of 84 km$^2$ at the stream gauging station (Figure 1). Its topography is comprised of mountainous areas (Mt. Elgon) from where the mainstream (Atari River) originates and flows to the gently rolling plains. From ASTER GDEM [40], its elevation range is 2389 m. Some 35 km$^2$ (42%) is forest, 28 km$^2$ (33%) is agricultural area and 21 km$^2$ (25%) is rangeland [41].

Under "the Project on Irrigation Scheme Development in Central and Eastern Uganda", hydro-meteorological monitoring equipment were set up in Atari River catchment in 2015, viz.: a mid-stream rain-gauge to detect catchment rainfall; a downstream meteorological station to measure weather parameters required for the calculation of evapotranspiration (ET$_0$) using the FAO Pemman-Montieth method [42], i.e.,: wind speed, air temperature, humidity, and solar radiation; and a water level sensor at a control section of the Atari River. In addition, discharge measurements were conducted to obtain the rating curve for Atari River [7].

### 2.2. Tank Model

Tank model is a conceptual, lumped model that was proposed and developed by Sugawara et al. in the 1950s [43]. One version of the model comprises of four tanks laid out vertically in series, so named tanks 1, 2, 3 and 4 (Figure 2). Rainfall data is inputted to tank 1 while ET$_0$ is subtracted from it. If tank 1 is empty, ET$_0$ is deducted from tank 2. If both tanks 1 and 2 are empty, then ET$_0$ is subtracted from tank 3 and so on. The side outlets of tanks 1, 2, 3 and 4 release surface runoff, through flow, sub-base runoff and base flow respectively. Additionally, tanks 1, 2 and 3 have bottom outlets, through which infiltration to a lower tank occurs [20].

Tank model is calibrated to determine the value of 16 unknown parameters, i.e.,: Co-efficient of the top side out let of tank 1 (A$_1$); co-efficient of the lower side outlet of tank 1 (A$_2$); co-efficient of the side outlet of tank 2 (B$_1$); co-efficient of the side outlet of tank 3 (C$_1$); co-efficient of the side outlet of tank 4 (D$_1$); co-efficient of the bottom outlet of tank 1 (A$_0$); co-efficient of the bottom outlet of tank 2 (B$_0$); co-efficient of the bottom outlet of tank 3 (C$_0$); height of the top side outlet of tank 1 (AH$_1$); height of the lower side outlet of tank 1 (AH$_2$); height of the side outlet of tank 2 (BH); height of the side outlet of tank 3

(CH); initial height of water in tank 1 (SA$_0$); initial height of water in tank 2 (SB$_0$); initial height of water in tank 3 (SC$_0$); and the initial height of water in tank 4 (SD$_0$), as in Figure 2.

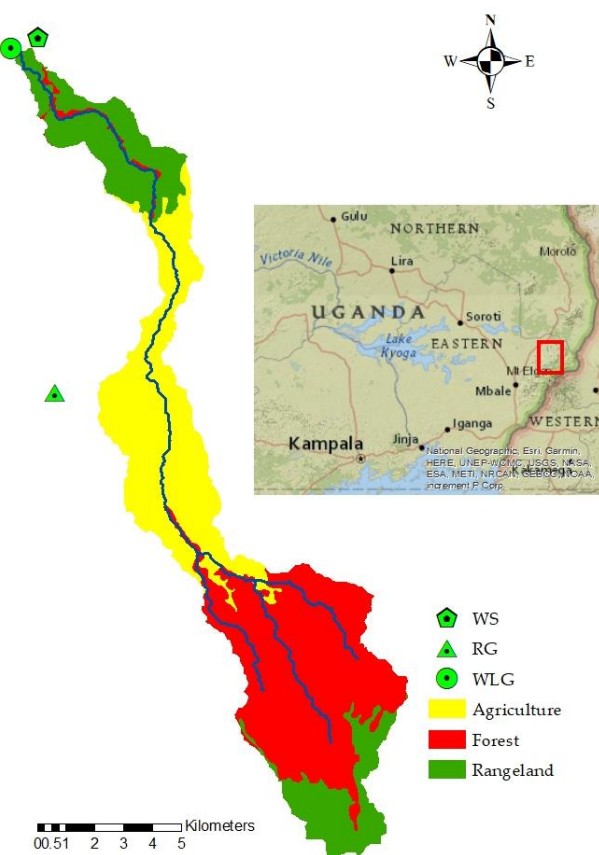

**Figure 1.** Instrumentation and land use in Atari River catchment. WS (weather station); RG (rain gauge); WLG (water level gauge).

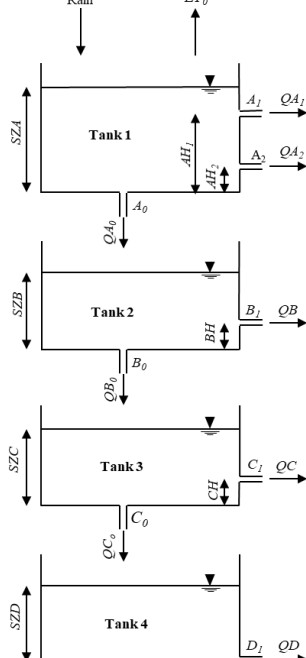

**Figure 2.** Schematic of tank model [25].

The parameters of tank model are not directly physical in nature. Rather, they can be related to some physical catchment characteristics: the co-efficients of the side and bottom outlets can be related to soil permeability in the horizontal and vertical directions respectively while the levels of the tanks from top to bottom are analogous to various depths within the soil layer.

The simulated river discharge is the sum of runoff from all the side outlets. Details of tank model are in [20].

### 2.3. TOPMODEL

TOPMODEL is a conceptual, semi-distributed model concept that was suggested by Beven and Kirkby [44]. Although a conceptual model, its formulation has some 'physically-meaningful' parameters [35,44]. The model separates the soil layer into a root zone, an unsaturated zone and a saturated zone. The analysis of the root zone and the unsaturated zone is done at grid-scale-the spatial resolution of the Digital Elevation Model (DEM)-, accounting for the spatial variability in catchment physical properties. Meanwhile, the saturated zone is computed at a lumped scale, with no consideration for the spatial variability of catchment physical properties. TOPMODEL evaluates the propensity of a soil to generate surface runoff from the Topographical Index (TI). The TI can be derived from a contour map or like in this study, a DEM [40]. A histogram of the TI distribution of Atari River catchment is shown in Figure 3.

$$\mathrm{TI} = \ln \frac{a_i}{\tan \beta_i} \tag{1}$$

where $a_i$ is the upstream contributing area per unit contour length, $\tan \beta_i$ is the local slope and $i$ is the grid number.

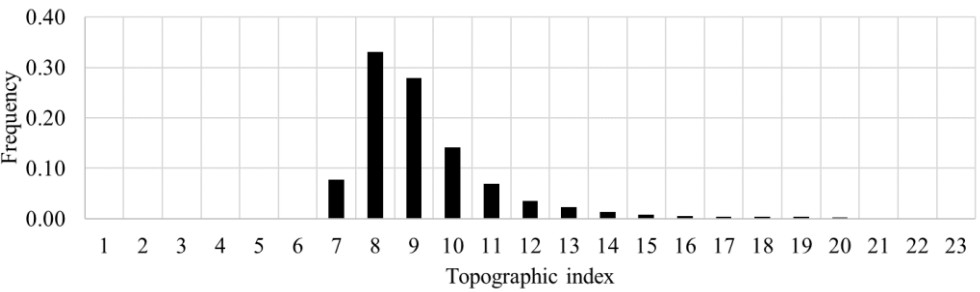

**Figure 3.** A histogram of the TI distribution of Atari River catchment.

TOPMODEL tracks the water balance in the root zone, the saturated zone and the unsaturated zone. The model was calibrated to find the values of 5 unknown parameters, i.e.,: exponential decay parameter ($m$); downslope saturated transmissivity ($T_e$); delay time constant ($t_d$); maximum root zone storage deficit ($SRZ_{max}$) and the initial root zone storage deficit ($SRZ_{initial}$).

The parameters of TOPMODEL are more directly related to the physical environment and could be obtained by measurement or calculation. Parameter $m$ could be estimated by analysing the baseflow recession curve [35], while $T_e$ can be measured [45].

A schematic of the TOPMODEL concept is shown in Figure 4. The TOPMODEL version described in Mukae et al. [46] was applied in this study. Details of TOPMODEL are in Beven, Beven and Kirkby and Mukae et al. [8,44,46,47].

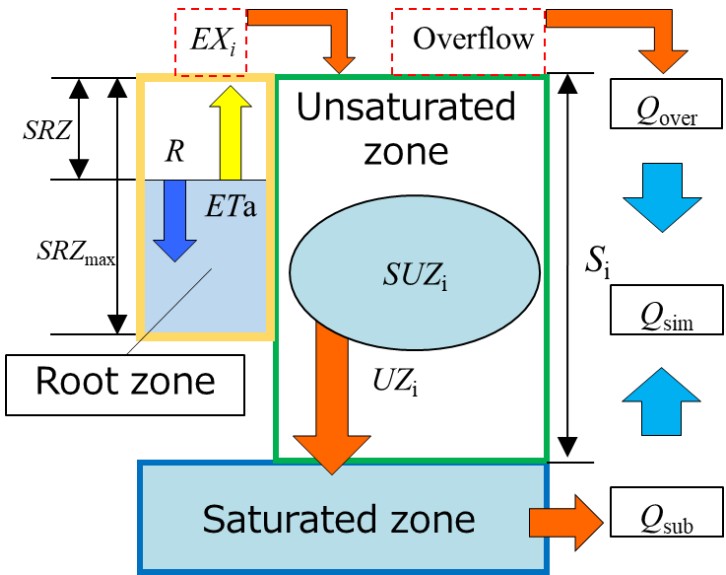

**Figure 4.** Schematic of the TOPMODEL concept [46]. SRZ (storage in the rootzone); $SRZ_{max}$ (maximum rootzone storage deficit); $S_i$ (local storage deficit); $SUZ_i$ (storage in the unsaturated zone); $UZ_i$ (drainage flux); $Q_{sim}$ (simulated discharge); $Q_{over}$ (overland flow); $Q_{sub}$ (subsurface flow).

*2.4. Data Requirement for Tank Model and TOPMODEL*

The input data for the models was observed rainfall, observed stream discharge and $ET_0$. Satellite data products have the potential to provide finer spatial resolutions of rainfall. Indeed, Kobayashi et al. [48] have explored the accuracy of 'Global Satellite Mapping of Precipitation (GSMaP)' satellite rainfall data products in Eastern Uganda and found their detection accuracy to be promising. However, Takido et al. [49] raised the concern of the inaccuracies of satellite rainfall products in high-elevation areas. Recognising that Atari River catchment has high-elevation areas, the study by Kobayashi et al. [48] needs furthering to include the verification of the accuracy of GSMaP satellite rainfall products in the higher elevation areas of Eastern Uganda.

Tropical rainfall is highly localised, and there is a need for multiple spatially distributed rain gauge networks to get more meaningful representations of catchment rainfall [50]. However, given the acceptable response of the observed downstream discharge to the rainfall recorded by the midstream-elevation rain gauge [7], it (mid-elevation rain gauge), at an elevation of 1961 m above sea level, was used to represent catchment-scale rainfall.

Rainfall was measured for each event and other hydro-meteorological parameters were recorded at 10-min logging intervals, with the daily averages being used for computation. $ET_0$ was calculated by inputting the weather parameters from the downstream rain gauge into the FAO Penman equation [51].

Sorooshian et al. [52] found 1-year data to be sufficient for the calibration of the soil moisture accounting model of the U.S. National Weather Service's river forecast system (SMA-NWSRFS). Further, because of a reduction in the marginal benefit of longer data periods, 2 or 3 years of the right kind of data were thought to be sufficient for the calibration period [53]. In Yapo et al. [54], about 8 years of data were required to calibrate a flood forecasting model. Since the study required the forecasting of a continuous runoff hydrograph, about 1 year of calibration data was considered sufficient.

Observed data for 1 March 2015 to 31 December 2015, and 1 January 2016 to 29 June 2016 were used for model calibration and validation respectively. Correspondingly, the total amount of rainfall, evapotranspiration and river discharge during the observation period in each year were 1655 mm; 1376 mm; and 534 for the year 2015 and 752 mm; 747 mm and 312 mm for the year 2016.

### 2.5. Calibration, Validation and Evaluation of Tank Model and TOPMODEL Efficiency

Stefnisdóttir et al. [55] evaluated the values of 3 metaheuristics for the calibration of the Hydrologiska Byråns Vattenbalansavdelning (HBV) model, namely: Monte Carlo (MC), Simulated Annealing, and Genetic Algorithm. Due to its simplicity and ability to identify good parameter sets with sufficient model iterations [55], the MC calibration procedure, performed using the Python Programming Language, was used for this study.

Calibration by the MC method was done by randomly selecting parameter values fitting within predefined maximum and minimum bounds following a uniform distribution. The parameters values were then ranked based on the Nash and Sutcliffe Efficiency (NSE) [56]. Foglia et al. [57] classified NSE values as: insufficient; sufficient; good; very good and excellent for NSE values of: <0.2; 0.2–0.4; 0.4–0.6; 0.6–0.8; and >0.8 respectively.

An additional objective function, 'the RMSE-observations standard deviation ratio (RSR)' [58], was used. RSR standardises the Root Mean Square Error (RMSE) value, enabling the judgement on the non-rejection or rejection of hydrological model simulations. Moriasi et al. [58] classified RSR values in the ranges of: $0.00 < RSR < 0.50$; $0.50 < RSR < 0.60$; $0.60 < RSR < 0.70$ and $RSR > 0.70$ as very good; good; satisfactory and unsatisfactory respectively. In Kastridis et al. [59], RSR values close to 0.5 were taken to represent an acceptable model prediction.

$$\text{NSE} = 1 - \left( \frac{\sum_1^n (\text{observed value} - \text{simulated value})^2}{\sum_1^n (\text{observed value} - \text{mean observed value})^2} \right) \tag{2}$$

$$\text{RMSE} = \sqrt{\frac{\sum_1^n (\text{observed value} - \text{simulated value})^2}{n}} \tag{3}$$

$$\text{STD}_{\text{obs}} = \sqrt{\frac{\sum_1^n (\text{observed value} - \text{mean of observed values})^2}{n}} \tag{4}$$

$$\text{RSR} = \frac{\text{RMSE}}{\text{STD}_{\text{obs}}} \tag{5}$$

where n is the number of observation days and $\text{STD}_{\text{obs}}$ is the standard deviation of the observed discharge.

To test the ability of the models to predict hydrographs, validation was carried out in the year 2016.

## 3. Results

### 3.1. MC Calibration Procedure

Using the MC calibration procedure; 10,000,000 iterations yielded the best performing parameter sets for both tank model and TOPMODEL: the 10,000,000 iterations were the maximum possible considering computer constraints. Therefore, all the parameter sets discussed below are drawn from 10,000,000 model iterations.

### 3.2. TOPMODEL

The best performing parameter sets from model calibration—see Table 1—were classified as 'very good' and 'satisfactory' in terms of NSE and RSR values respectively.

**Table 1.** The best performing parameter sets of the Atari River catchment during the calibration period.

| Year | $m$ (mm) | $T_e \times 10^9$ (mm²/day) | $t_d$ (day/mm) | $SRZ_{initial}$ (mm) | $SRZ_{max}$ (mm) | NSE | RSR |
|------|----------|------------------------------|-----------------|----------------------|-------------------|------|------|
| 2015 | 30 | 9.510 | 0.008 | 0.100 | 0.020 | 0.631 | 0.608 |

Table 2 shows that during model calibration, there was variability among parameter sets: equifinality within a similar calibration period. This dynamic nature of the parameters caused uncertainty about their physical meaning.

**Table 2.** Descriptive statistics of TOPMODEL parameter sets that were obtained during model calibration. Only parameter sets with an NSE of at least 0.5 were considered.

| Year | | $m$ (mm) | $T_e \times 10^9$ (mm²/day) | $t_d$ (day/mm) | $SRZ_{initial}$ (mm) | $SRZ_{max}$ (mm) |
|---|---|---|---|---|---|---|
| | Maximum | 50 | 10.000 | 0.020 | 0.100 | 0.100 |
| | Minimum | 20 | 0.002 | 0.004 | 0.100 | 0.002 |
| 2015 | Mean | 32 | 4.790 | 0.009 | 0.100 | 0.051 |
| | STD [+] | 8 | 2.920 | 0.003 | 0.000 | 0.028 |
| | Cov * (%) | 25.361 | 60.969 | 33.718 | 0.000 | 54.992 |

[+] Standard deviation; * Co-efficient of variance.

The validation of TOPMODEL yielded an NSE value of 0.677 (classified as very good) and an RSR of 0.568 (classified as good): Table 3. The model better reproduced the hydrograph for 2016 than that for 2015.

**Table 3.** A comparative performance of tank model and TOPMODEL during model calibration and validation periods. Years 2015 and 2016 were used for model calibration and validation respectively.

| | Tank Model | | | | TOPMODEL | | | |
|---|---|---|---|---|---|---|---|---|
| | Calibration | | Validation | | Calibration | | Validation | |
| Year | NSE | RSR | NSE | RSR | NSE | RSR | NSE | RSR |
| 2015 | 0.737 | 0.513 | | | 0.631 | 0.608 | | |
| 2016 | | | 0.396 | 0.778 | | | 0.677 | 0.568 |

Generally, there was a similar trend between the observed and simulated hydrographs, with an observable response of the simulated hydrograph to rainfall input (refer to Figure 5).

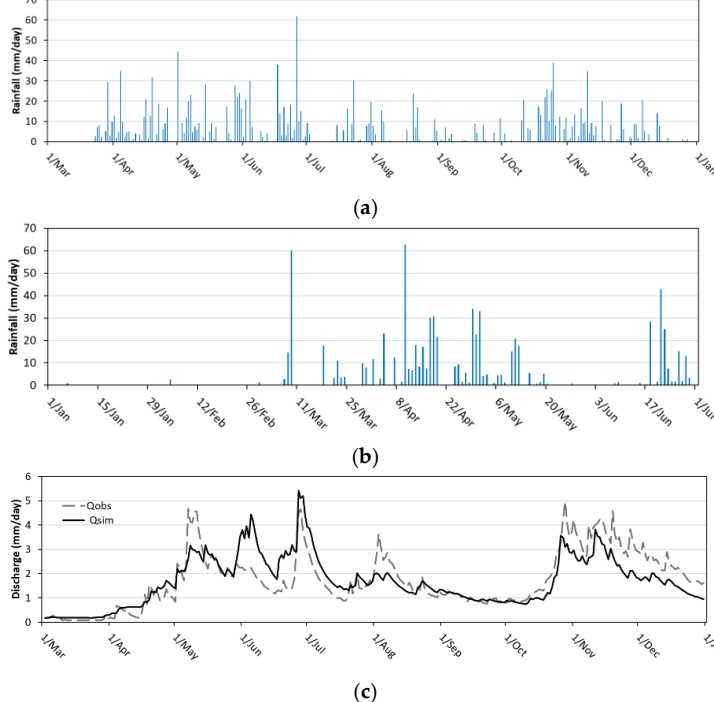

**(a)**

**(b)**

**(c)**

**Figure 5.** *Cont.*

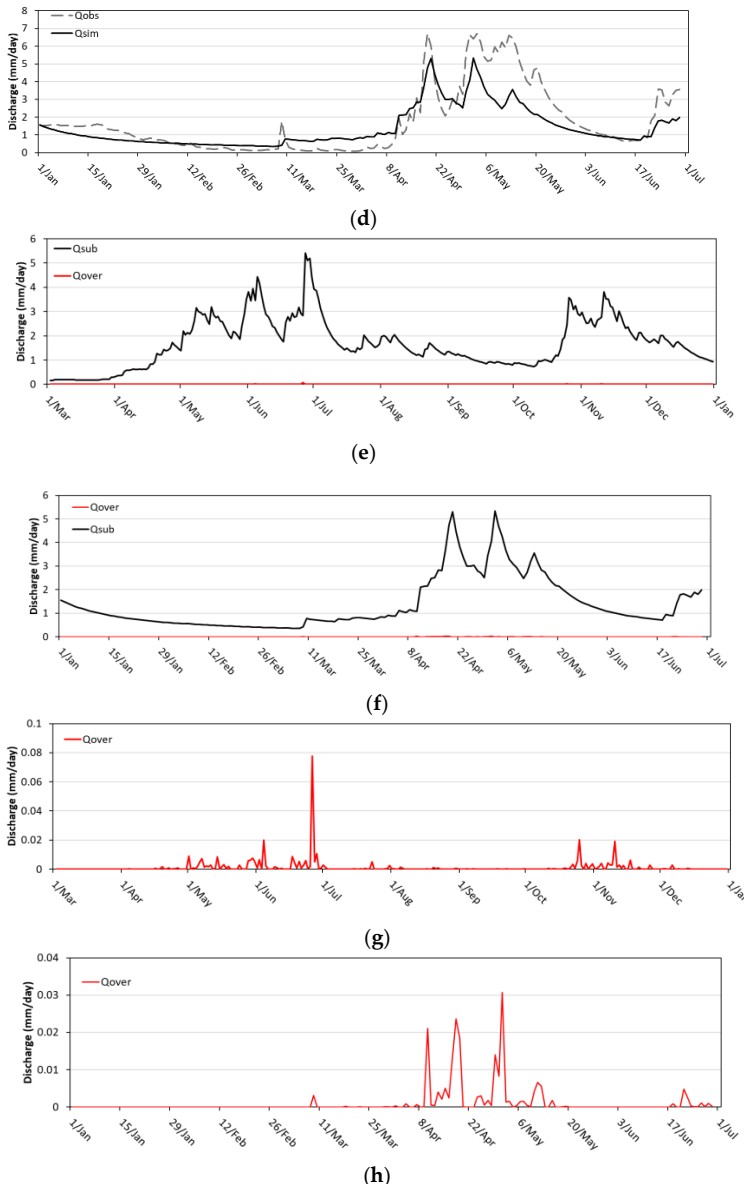

**Figure 5.** Observed rainfall, observed hydrograph and the hydrographs simulated by TOPMODEL. (**a**) Observed rainfall in 2015; (**b**) Observed rainfall in 2016; (**c**) Observed hydrograph and the hydrograph simulated during model calibration in 2015; (**d**) Observed hydrograph and the hydrograph simulated during model validation in 2016; (**e**) Hydrograph separation in 2015 during model calibration. Note that the overland flow was too small to be represented on this graph; (**f**) Hydrograph separation in 2016 during model validation. Note again that the overland flow was too small to be detected on this graph; (**g**) Overland flow prediction in 2015 during model calibration; and (**h**) Overland flow prediction in 2016 during the model validation period. $Q_{obs}$ (observed discharge); $Q_{sim}$ (simulated discharge); $Q_{over}$ (overland flow); $Q_{sub}$ (subsurface flow).

From Figure 6, all parameters–except $m$ and $t_d$–showed good (and bad) simulations over the whole range of the parameter space. As the NSE value increased, $m$ and $t_d$ occupied a narrower zone in the parameter space.

### 3.3. Tank Model

The best performing parameter sets following model calibration were classified as 'very good' and 'good' in terms of NSE and RSR objective functions respectively (Table 4).

Model validation yielded an NSE value of 0.396 (classified as a good fit) and an RSR of 0.778 (classified as unsatisfactory): Table 3.

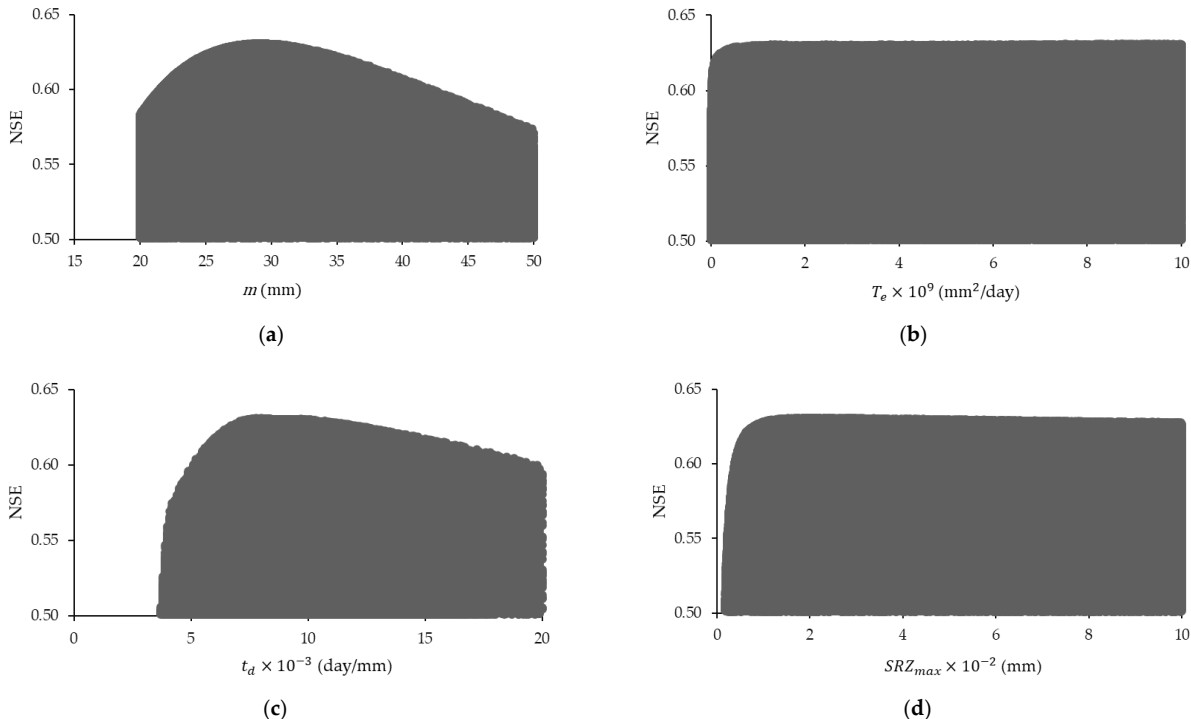

**Figure 6.** Scatter plots of NSE values versus the corresponding parameter values for TOPMODEL parameter sets that were got during the model calibration period. Each dot represents a separate parameter value from a given parameter set. Only simulations with an NSE value of at least 0.5 are shown. (**a**) A scatter plot of NSE versus values of parameter $m$; (**b**) A scatter plot of NSE versus values of parameter $T_e$; (**c**) A scatter plot of NSE versus values of parameter $t_d$; (**d**) A scatter plot of NSE versus values of parameter $SRZ_{max}$.

**Table 4.** The best performing tank model parameter sets for Atari River catchment during model calibration in 2015.

| Year | $A_1$ | $A_2$ | $A_0$ | $B_1$ | $B_0$ | $C_1$ | $C_0$ | $D_1$ | $SA_0$ (mm) | $SB_0$ (mm) | $SC_0$ (mm) | $SD_0$ (mm) | $AH_1$ (mm) | $AH_2$ (mm) | $BH$ (mm) | $CH$ (mm) | NSE | RSR |
|------|-------|-------|-------|-------|-------|-------|-------|-------|------|------|------|------|------|------|------|------|-----|-----|
| 2015 | 0.026 | 0.020 | 0.011 | 0.017 | 0.003 | 0.014 | 0.001 | 0.003 | 79 | 926 | 100 | 19 | 508 | 26 | 3880 | 392 | 0.737 | 0.513 |

From Table 5, it was evident that during the calibration period, there was variability among parameter sets: same calibration period equifinality. This caused uncertainty about the physical meaning of the parameters.

**Table 5.** Descriptive statistics for tank model parameter sets that were obtained from model calibration. Only parameter sets with an NSE of at least 0.5 were considered.

| Year | | $A_1$ | $A_2$ | $A_0$ | $B_1$ | $B_0$ | $C_1$ | $C_0$ | $D_1$ | $SA_0$ (mm) | $SB_0$ (mm) | $SC_0$ (mm) | $SD_0$ (mm) | $AH_1$ (mm) | $AH_2$ (mm) | $BH$ (mm) | $CH$ (mm) |
|------|---------|-------|-------|-------|-------|-------|-------|-------|-------|------|------|------|------|------|------|------|------|
|      | Maximum | 0.100 | 0.075 | 0.064 | 0.069 | 0.054 | 0.060 | 0.027 | 0.042 | 100 | 1000 | 100 | 100 | 1000 | 100 | 10000 | 10000 |
|      | Minimum | 0.007 | 0.006 | 0.003 | 0.001 | 0.001 | 0.001 | 0.000 | 0.000 | 10 | 192 | 100 | 0 | 1.587 | 0.002 | 166 | 0.641 |
| 2015 | Average | 0.055 | 0.027 | 0.019 | 0.016 | 0.011 | 0.009 | 0.007 | 0.005 | 55 | 740 | 100 | 47 | 564 | 40 | 5290 | 5060 |
|      | STD     | 0.022 | 0.009 | 0.008 | 0.008 | 0.005 | 0.005 | 0.003 | 0.003 | 26 | 175 | 0 | 29 | 253 | 24 | 2718 | 2855 |
|      | CoV (%) | 41 | 34 | 42 | 51 | 46 | 61 | 51 | 61 | 46 | 24 | 0 | 60 | 45 | 59 | 51 | 56 |

Like TOPMODEL, there was generally a similar trend between the observed and the simulated hydrograph during the wet season. However, the hydrograph response was not sensitive to the small rainfall events after a dry spell (refer to Figure 7).

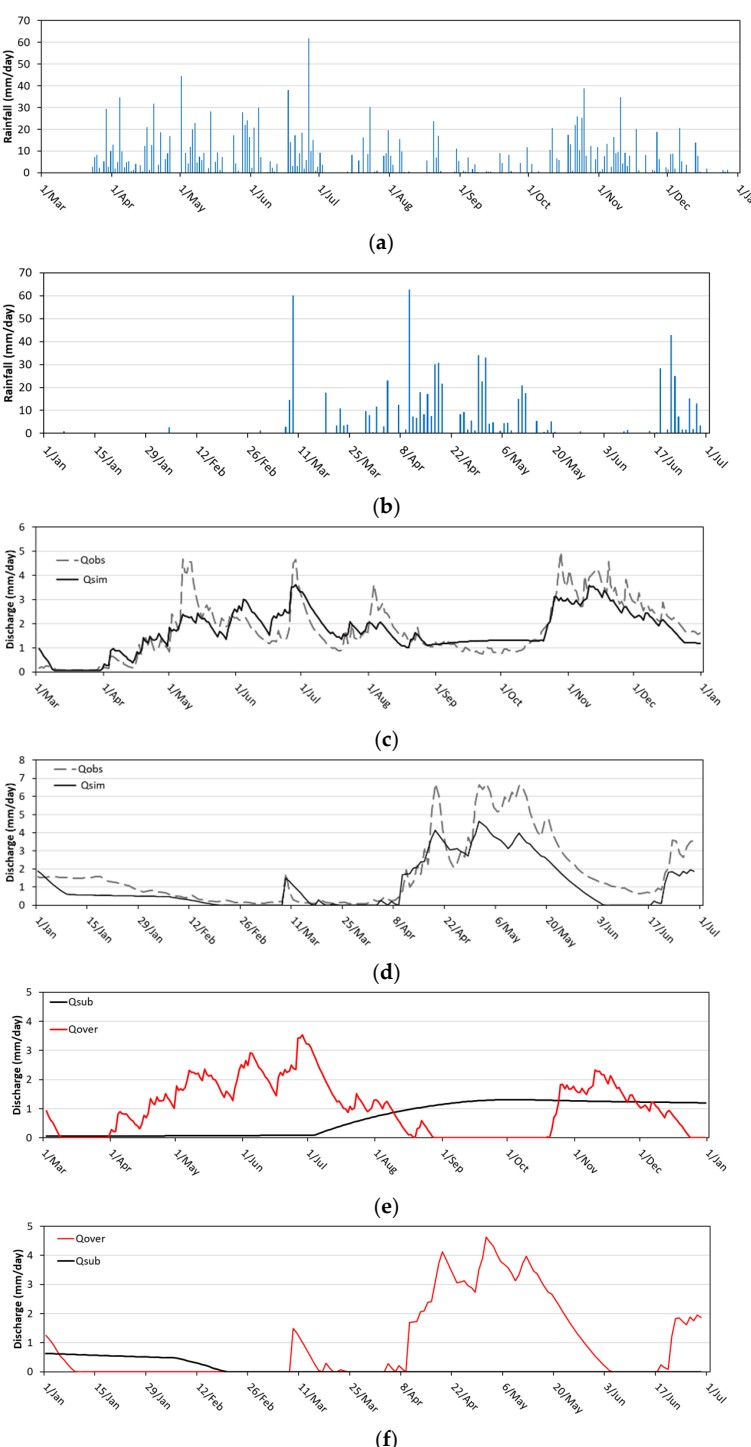

**Figure 7.** Observed rainfall, observed hydrograph and the hydrographs simulated by tank model. (**a**) Observed rainfall in 2015; (**b**) Observed rainfall in 2016; (**c**) Observed hydrograph and the hydrograph simulated during model calibration in 2015; (**d**) Observed hydrograph and the hydrograph simulated during model validation in 2016; (**e**) Hydrograph separation in 2015 during model calibration; (**f**) Hydrograph separation in 2016 during model validation. $Q_{obs}$ (observed discharge); $Q_{sim}$ (simulated discharge); $Q_{over}$ (overland flow); $Q_{sub}$ (subsurface flow).

From Figure 8, as the NSE value increased, all the parameters-except $A_1$; $AH_1$; $BH$ and $CH$–occupied a narrower zone in the parameter space.

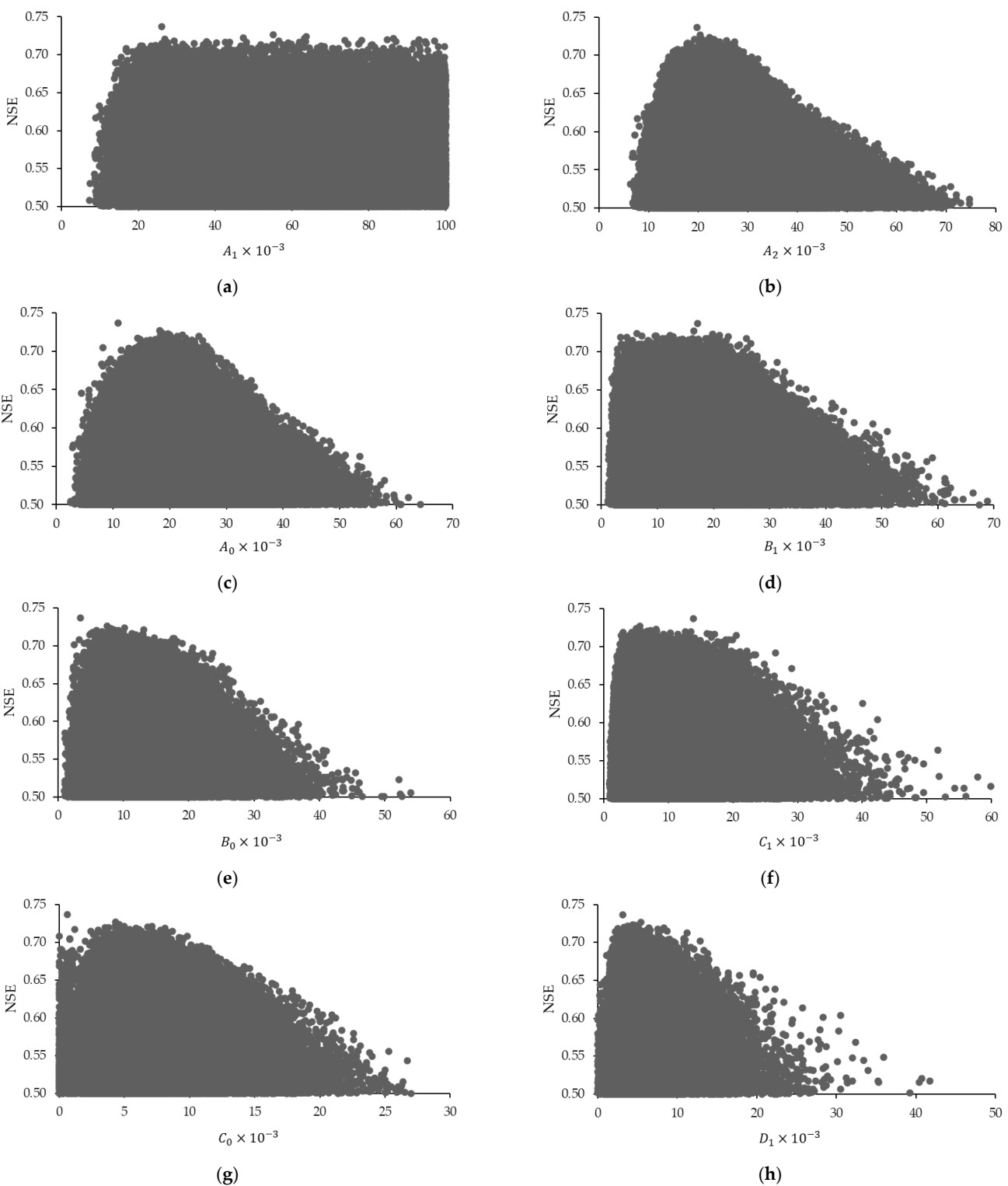

**Figure 8.** *Cont.*

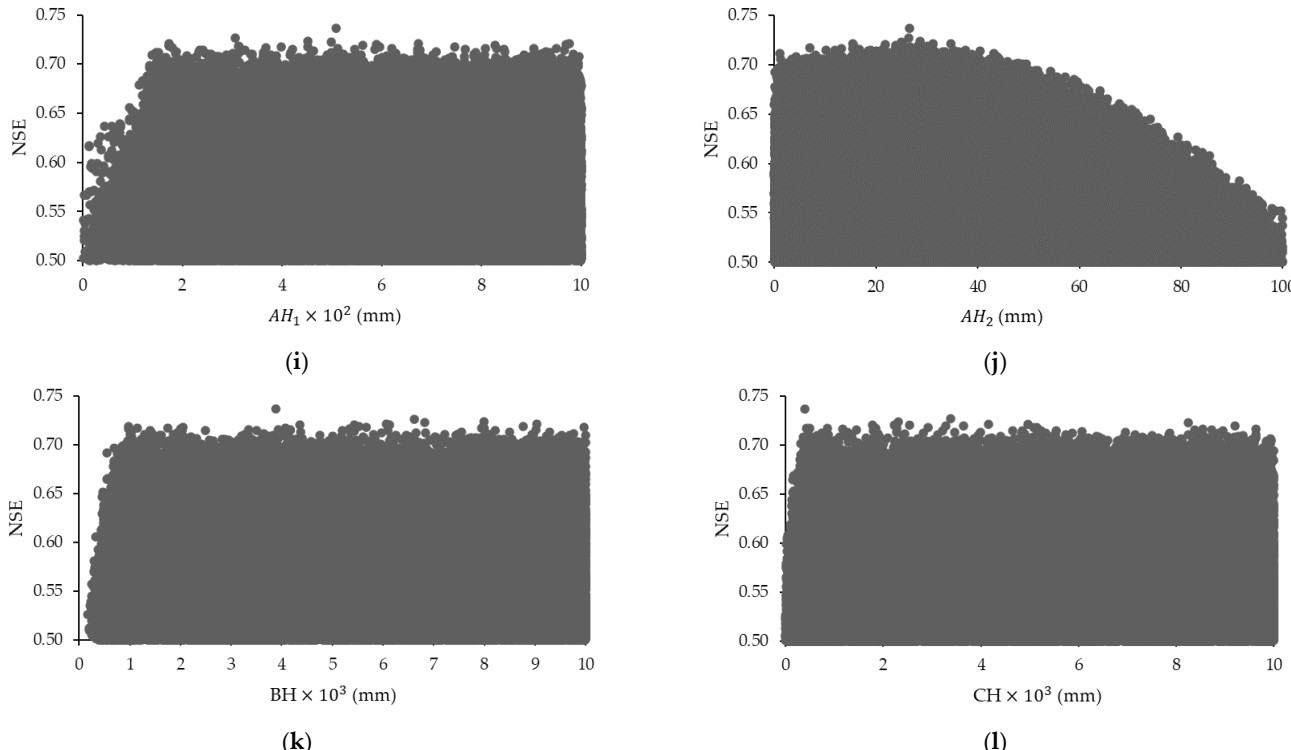

**Figure 8.** Scatter plots of NSE values versus the corresponding parameter values for tank model parameter sets that were got during model calibration. Each dot represents a separate parameter value from a given parameter set. Only simulations with an NSE value of at least 0.5 are shown. (**a**) A scatter plot of NSE versus values of parameter $A_1$; (**b**) A scatter plot of NSE versus values of parameter $A_2$; (**c**) A scatter plot of NSE versus values of parameter $A_0$; (**d**) A scatter plot of NSE versus values of parameter $B_1$; (**e**) A scatter plot of NSE versus values of parameter $B_0$; (**f**) A scatter plot of NSE versus values of parameter $C_1$; (**g**) A scatter plot of NSE versus values of parameter $C_0$; (**h**) A scatter plot of NSE versus values of parameter $D_1$; (**i**) A scatter plot of NSE versus values of parameter $AH_1$; (**j**) A scatter plot of NSE versus values of parameter $AH_2$; (**k**) A scatter plot of NSE versus values of parameter $BH$; (**l**) A scatter plot of NSE versus values of parameter $CH$.

### 3.4. Comparison between Tank Model and TOPMODEL

From Table 3, it was apparent that during the calibration period, both models had 'very good' performances in terms of the NSE objective function values. In terms of RSR values, tank model and TOPMODEL had 'good' and 'satisfactory' performances respectively. It follows that tank model had a better objective value performance than TOPMODEL during the calibration period.

However, during validation, TOPMODEL outperformed tank model, with TOP-MODEL having a 'very good' NSE performance, and a 'good' RSR performance while tank model had a 'sufficient' NSE performance and an 'unsatisfactory' RSR performance.

As depicted in Figures 6 and 8, during model calibration, equifinality in the same calibration period was observed. Given the dynamic nature of the parameters, uncertainty in the physical meaning of the TOPMODEL and tank model parameters crept in.

From Figures 5 and 7, the response of the simulated hydrographs was generally in tandem with that of the observed hydrograph, albeit with under estimation or sometimes overestimation of discharge. However, during a low rainfall period-e.g., between September and November of 2015-the simulated hydrograph pattern due to tank model did not mirror that of the observed hydrograph. On the other hand, in the same period, TOPMODEL fared better in mirroring the observed hydrograph pattern.

During a rainfall event(s), much of the runoff predicted by TOPMODEL was subsurface flow while tank model predicted most of the flow as overland flow.

## 4. Discussion

With increased computing power, it was possible to run up to 10,000,000 model iterations during calibration. As the number of iterations increased, so did the chances of getting parameter sets with high calibration efficiency performances. However, this heightened equifinality, with many different parameter sets having equal predictive performance based on an objective function. This then calls on the modeler to 'optimise' parameter set decisions based on experience, perceived responses of the catchment and the shape of the baseflow curve.

The value of meaningful observed data in the rejection of competing parameter sets cannot be overlooked [60]. Thanks to equifinality, the modeler is reminded of the need to have a personal perception of the physical world, especially through field investigations, prompting a return to classical hydrology, as the search for parameter sets that ought to be rejected continues.

Satellite rainfall products offer gridded, finer spatial resolution data for running hydrological models. Kobayashi et al. [48] found as acceptable the detection accuracy of GSMaP satellite rainfall products in Eastern Uganda. However, their research focussed on the low to mid-elevation areas. Indeed, satellite rainfall products have been found to be inaccurate in the higher elevation areas [49]. Therefore, the study by Kobayashi et al. [48] needs furthering to include the verification of the accuracy of GSMaP satellite rainfall products in higher elevation areas in Eastern Uganda. When this is clarified, then satellite rainfall products would be considered as input data for hydrological models in Eastern Uganda.

It is hypothesised that tank model had a more pronounced equifinality than TOPMODEL. This premise is based on the reasoning that because of its greater number of calibrated parameters (16), their (parameter) interdependence was greater than that among the 5 TOPMODEL parameters. If this were the case, then it can be further argued that compared to tank model, TOPMODEL parameters had a stronger relationship with the physical characteristics of the catchment.

In being able to respond to the low rainfall events during/after a dry period, TOPMODEL showed a more robust performance especially in the wetting up phase after the localisation and disconnection of saturated areas. This robustness could be attributed to the fact that TOPMODEL is built on the 'variable source area' concept [44]. On the other hand, tank model, which assumes a lumped rainfall-runoff response, was unable to respond meaningfully during the wetting up phase following disconnection of saturated areas. The challenges of model response when localisation and separation of saturated areas occurs are elucidated in [61].

The 'over-prediction' of flow as subsurface flow and the subsequent 'under-prediction' of overland flow by TOPMODEL following a rainfall event(s) could be attributed to the large value of the $T_e$ parameter. This could point to the existence of the 'preferential' flow pathways in the catchment that are described in Beven [61]. The differing behaviour of both models in hydrograph separation during rain season(s) requires further investigation, e.g., through hydrograph separation experiments, etc., to affirm the more realistic flow separation curves. Or perhaps the details of hydrograph separation are not important for irrigation planning?

Cases of under prediction of peak discharge could be attributed to localised rainfall events that were missed by the rain gauge. Indeed, Sugawara [50] reported that tropical rainfall was highly localised, requiring multiple spatially distributed rain gauge networks to get a more meaningful representation of catchment rainfall. When the shortcomings of GSMaP rainfall products identified by Takido et al. [49] are clarified, they (GSMaP products) could complement ground observed data, offering finer spatial resolutions.

The superior performance of TOPMODEL during validation could be attributed to the fact that TOPMODEL parameters were more physically meaningful than those of tank model. Indeed, some TOPMODEL parameters like $m$, and $T_e$ can be indirectly calculated or measured directly: an estimation of parameter $m$, for example, can be found by the analysis of the baseflow recession curve [35] while parameter $T_e$ can be measured directly [45].

Additionally, cognisant of the equifinality phenomenon, it is possible that we were 'lucky' enough to find a TOPMODEL parameter set that performed well both in calibration and validation, and that of tank model is hidden somewhere in the parameter space, waiting to be found.

Based on acceptable NSE and RSR values during model calibration, both tank model and TOPMODEL were applicable to Atari River catchment for the simulation of daily river discharge for irrigation and drainage planning. However, owing to its better performance during the validation period, TOPMODEL was found to be a more robust model for run-off forecasting in Atari River catchment. This showed that TOPMODEL, with its more complex model formulation, and the more physically meaningful parameters, was superior in predicting discharge during a 'data-less' period. This then justified, to an extent, the recent drive towards the more process-based models.

However, in no way was this an endorsement of TOPMODEL's superiority over tank model, but rather an admission that the authors were able to find a TOPMODEL parameter set that performed well both in calibration and validation: a tank model parameter set that could match or even exceed TOPMODEL's validation performance could exist somewhere in the parameter space, waiting to be discovered. Indeed, owing to its larger number of unknown parameters (16), equifinality in tank model might be exacerbated, going by Beven's reasoning [60].

TOPMODEL also had the added benefit of having fewer parameters (5 parameters) that required calibration than tank model (16 parameters). Moreover, TOPMODEL could represent the soil moisture content at a spatially distributed scale which can have important applications in irrigation scheduling or drainage design. Okiria et al. [39] also reported the possibility of using a well calibrated TOPMODEL as a tool to detect errors in observed hydro-meteorological data, given the physical meaning of its parameters. Nevertheless, the challenge of estimating the initial conditions of TOPMODEL must be addressed to ensure its successful application for irrigation system design and operation.

## 5. Conclusions

Cognisant of the recent drive towards the more complex, process-based models, there ought to be a justification for preferring complexity over simplicity. This study attempted to tackle this issue by comparing the simpler tank model as a lumped model with the more complex TOPMODEL as a semi-distributed model. The study was done for the Atari River catchment, in Eastern Uganda. During model calibration, the performance of both models was closely matched in terms of NSE and RSR values. Owing to TOPMODEL's better performance during model validation (a 'very good' NSE value and a 'good' RSR value for TOPMODEL compared to a 'sufficient' NSE value and an 'unsatisfactory' RSR value for tank model), TOPMODEL was judged to be better at forecasting hydrographs in Atari River catchment. This showed that the more complex model formulation, with the more physically meaningful parameters–TOPMODEL–showed robustness in predicting discharge during a 'data-less' period. This then justified the recent drive towards the more process-based models. TOPMODEL's strengths can be further enhanced by addressing the challenges of estimating the initial conditions. Going forward, more meaningful data–observed rainfall and discharge, etc.-, a more rigorous calibration regime, and an expanded study, are needed to confirm the findings herein.

**Author Contributions:** Conceptualisation, E.O.; methodology, E.O.; software, E.O, H.O. and Y.Y.; validation, H.O., S.S. and K.N.; formal analysis, E.O.; investigation, E.O.; data curation, Y.K. and E.O.; writing—original draft preparation, E.O.; writing—review and editing, H.O.; K.N. and Y.K.; visualization, E.O.; supervision, H.O.; S.S. and K.N.; project administration, H.O. and K.N. All authors have read and agreed to the published version of the manuscript.

**Funding:** This work was financially supported by JST SPRING, Grant Number JPMJSP2125.

**Data Availability Statement:** Restrictions apply to the availability of these data. Data were obtained from MAAIF and MWE and are available on request from the corresponding author with permission from MAAIF and MWE.

**Acknowledgments:** Appreciation goes to the offices of the District Agricultural Engineers (DAEs) of Bulambuli, Kween, Mbale and Sironko Districts for the help in data collection. The support of Eng. Rajahb Namakhola, DAE, Mbale District for availing office space during field work in Uganda is highly appreciated. The author (E.O.) would like to take this opportunity to thank the "Interdisciplinary Frontier Next-Generation Researcher Program of the Tokai Higher Education and Research System".

**Conflicts of Interest:** The authors declare no conflict of interest.

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
