# Peer review of "A Comparative Evaluation of Lumped and Semi-Distributed Conceptual Hydrological Models: Does Model Complexity Enhance Hydrograph Prediction?"

_hydrology, doi:10.3390/hydrology9050089_

Round 1
Reviewer 1 Report
I enjoyed the key research idea of the paper and I would recommend if for publishing after major revisions. My comments as below:
Abstract: Can be improved providing. In its current format is too generic missing the key point of the work done. As the abstract is the first contact point of a reader I would use less reference of the type of models used and I would highlight with simple words the key research issue of the paper.
Introduction: The problem of modelling in data scarce area is not a local Ugandas issue. The introduction can be improved summarizing better the modelling approaches from data scarce to data rich areas globally. Then the authors could raise up the key idea of the paper.
Materials and methods: Missing a discussion on how they set up (interpret) the hydro logical response unit through the different approaches. I would assume that through the comparison of the two models the physical-base calibrated hydro logical parameters should be discussed
Figure 4: Poor image quality,
Two Figures 4 please check the numbering. Second figure 4 graphs are invisible.
Discussion: I think missing a discussion of global available remote sensing for using hydro logical calibration in data
Conclusion: This section must be expanded providing more details.
Reviewer 2 Report
Dear Editor.
I have finished my review on the proposed paper “A comparative evaluation of lumped and semi-distributed conceptual hydrological models in a medium catchment: does model complexity enhance hydrograph prediction?” hydrology-1684707-peer-review-v1.
Summary of the manuscript:
In the proposed paper, the authors’ goal is to evaluate the performance of two hydrological models, tank model and the TOPMODEL, in order to find if the models with more parameters are predicting the flow discharge better than the simpler models. They found that TOPMODEL could predict the river floe discharge in higher accuracy than the tank model.
General review:
- Generally, the manuscript presents an interesting topic and the specific research seems to include some significant points for the research community of this field.
- The proposed paper is very well written with very good use of English language. Except some very minor grammatical mistakes and word errors, this paper is written with a very good scientific style. The authors should check again the paper to correct these minor mistakes.
- The proposed paper is very well structured. It begins with an analytical Introduction with the appropriate references that helps the reader to get into the subject immediately. In Introduction there is an effort to provide previous studies with similar scientific content, which took place in the research area and in other countries. Authors describe and set very well the scientific problem and how other researchers have approached. At the end of Introduction, authors clearly state the goals of the research. However, I think that authors could add more literature in Introduction from other countries.
- The methodology is generally very interesting, and well explained, so other researchers could easily repeat it. Every aspect of methodology is well documented with the use of the appropriate literature. However, I have some concerns about the RMSE. See below specific comments
- The results scientifically explained and are OK.
- The quality of the work in Discussion is generally OK.
- Conclusions are appropriate for this paper.
Additional points for revision:
In my opinion, the proposed paper could be characterized as a good research work, complies with aims of Hydrology.
Figure 1: As I can see there is no rain gauge in higher altitudes in the watershed. The watershed has very high altitudes and it is sure that the rain orographic effect is very intense in the study area. I suppose that most of the rain falls in the upper part of the watershed, in which there is no rain gauge. This is a huge disadvantage of the study and for the better calibration of the models. You should definitely discuss this limitation in the text. I wonder If it was better to use some satellite data for the upper part.
Lines 171-172: Eventually, which rain data did you use in hydrological modeling? From rain gauge or from the weather station?
Lines 187-190: You used the Nash and Sutcliffe Efficiency. The most used abbreviation for this statistic is NSE. Please, change it is all the text.
Line 190: You did not use literature about the RMSE. Also, you did not refer the optimal values of RMSE. RMSE describes the difference between observed and simulated data in the units of the variable. RMSE optimal value is zero (0) (Moriasi et al. 2007). Please, add the description with the provided literature.
Table 1, 3 and 4 – Results about the RMSE: You used the RMSE to evaluate the model performance. You present the results of RMSE in the tables. However, it is very surprising that in whole the text you did not discuss the results of RMSE. Are the RMSE values acceptable? In line 320 you say that RMSE values are acceptable for both models. How you support this finding?
We are not sure if these values of RMSE are acceptable. RMSE describes the difference between observed and simulated data in the units of the variable. RMSE optimal value is zero (0). For example, when the RMSE of a hydrological model in a river (with discharge values ranging between 200-600 m3/sec) is 2.292, could be considered very low and close to zero. However, in another case, for example temperature simulation model (with temperatures values ranging between 0.3 and 5.7 oC), an RMSE of 2.292 is huge.
So, the best way to interpret the RMSE and understand if a pairwise correlation is acceptable, is to provide the Standard Deviation (SD) of the observed data. It is known from previous studies that RMSE values less than half of the SD (Standard Deviation) of the observed data (RMSE/SDobs < 0.5) may be considered low and acceptable (Kastridis et al. 2021, Singh et al. 2005). Provide the SD (in a table) with the RMSE, discuss in the text the above mentioned, and explain if the RMSE values are acceptable in terms of the SD. Add the proposed (given) literature to support this discussion in the text.
References
Moriasi, D., et al. (2007). Model evaluation guidelines for systematic quantification of accuracy in watershed simulations. Transactions of the ASABE, 50(3), 885–900. https://doi.org/10.13031/2013.23153.
Kastridis, A. et al. 2021. Investigation of Flood Management and Mitigation Measures in Ungauged NATURA Protected Watersheds. Hydrology 2021, 8, 170. https://doi.org/10.3390/hydrology8040170.
Singh, J., et al (2005). Hydrological modeling of the iroquois river watershed using hspf and swat1. Journal of the American Water Resources Association, 41, 343–360. https://doi.org/10.1111/j.1752-1688.2005.tb03740.x.
Round 2
Reviewer 1 Report
I'm satisfied with author responses and I would recommend it for publishing as it is.
Reviewer 2 Report
Dear authors.
Thank you very much for your effort to improve your paper and for the responses to my comments. Your paper has significantly improved after the applied revisions. However, I have some minor comments to be added in the text.
- The shape of the watershed is very elongated. Taking into account the intense orographic effect of the area (highest altitude 3475 m.asl) I am sure that more rainfall falls in the higher altitudes. You say that: "The downstream discharge responded well to the rainfall captured by the mid-elevation rain gauge". Of course, I trust your analysis. However, the bad distribution of rain gauges is a well known limitation of the hydrological modeling. You can add a prase in the text about this limitation.
- The revised paper have a lot of corrections. You should be very careful when you will prepair the final version.
- line 64: "For example, in Uganda, the majority of rivers are ungauged [5]". The problem with the ungauged watersheds, river and stream is general and intense also in other countries. Do not limit your Introduction saying only about Uganda. Rephrase this sentence like below for example, adding the proposed literature from other countries: "The ungauged or poorly gauged watersheds are a common problem in scientific community (Sapountzis et al. 2021, Borga et al. 2010), and also in Uganda, where the majority of rivers are ungauged [5], a fact that increase the uncertainties in hydrological modelling".
References
Borga M. et al. (2010). Flash floods: Observations and analysis of hydro-meteorological controls. Journal of Hydrology, 394(1-2), 1–284.
Sapountzis M. et al. (2021), Utilization and uncertainties of satellite precipitation data in flash flood hydrological analysis in ungauged watersheds, Global NEST Journal, 23(3), 388-399.
